# Association of worsening of nonalcoholic fatty liver disease with cardiometabolic function and intestinal bacterial overgrowth: A cross-sectional study

**Marília Marques Pereira Lira**[1,2,3], **José Eymard Moraes de Medeiros Filho**[2,3], **Vinícius José Baccin Martins**[4], **Gitana da Silva**[3], **Francisco Antônio de Oliveira Junior**[4], **Éder Jackson Bezerra de Almeida Filho**[5], **Alexandre Sérgio Silva**[5], **João Henrique da Costa-Silva**[6], **José Luiz de Brito Alves**[1] *

1 Department of Nutrition, Health Sciences Center, Federal University of Paraiba, João Pessoa, Brazil, 2 Department of Internal Medicine, Medical Sciences Center, Federal University of Paraiba, João Pessoa, Brazil, 3 Lauro Wanderley Hospital, Federal University of Paraiba, João Pessoa, Brazil, 4 Department of Physiology and Pathology, Health Sciences Center, Federal University of Paraiba, João Pessoa, Brazil, 5 Department of Physical Education, Health Sciences Center, Federal University of Paraiba, João Pessoa, Brazil, 6 Department of Physical Education and Sport Sciences, Federal University of Pernambuco, Vitória de Santo Antão, PE, Brazil

* jose_luiz_61@hotmail.com, jose.luiz@academico.ufpb.br

**Data Availability Statement:** All relevant data are within the paper.

## Abstract

### Background & aims

Non-alcoholic fatty liver disease (NAFLD) has been associated with small bowel bacterial overgrowth (SIBO) and cardiometabolic dysfunction. This cross-sectional study aimed to evaluate the cardio-metabolic parameters and SIBO in patients with different degrees of hepatic fibrosis estimated by NAFLD fibrosis score (NFS).

### Methods

Subjects (n = 78) were allocated to three groups: Healthy control (n = 30), NAFLD with low risk of advanced fibrosis (NAFLD-LRAF, n = 17) and NAFLD with a high risk of advanced fibrosis (NAFLD-HRAF, n = 31). Anthropometrics, blood pressure, electrocardiogram and heart rate variability (HRV) were evaluated. Only the NAFLD-LRAF and NAFLD-HRAF groups were submitted to blood biochemical analysis and glucose hydrogen breath tests.

### Results

The NAFLD-HRAF group had higher age and body mass index when compared to the control and NAFLD-LRAF groups. The prevalence of SIBO in the NAFLD group was 8.33%. The low frequency/high-frequency ratio (LF/HF ratio) was augmented in NAFLD-LRAF ($p <$ 0.05) when compared with control group. NAFLD-HRAF group had a wide QRS complex ($p <$ 0.05) and reduced LF/HF ratio ($p <$ 0.05) compared to the control and NAFLD-LRAF groups. Serum levels of albumin and platelets were more reduced in the NAFLD-HRAF subjects ($p <$ 0.05) than in the NAFLD-LRAF.

**Funding:** The authors thank the Coordination for the Improvement of Higher Education Personnel (Coordenação de Aperfeiçoamento de Pessoal de Nível Superior- CAPES, Brazil) for financial aid to students support (code 0002) awarded to MMPL. The funders had no role in study design, data collection and analysis, decision to publish, or preparation of the manuscript.

**Competing interests:** The authors have declared that no competing interests exist.

## Conclusions

NAFLD impairs cardiac autonomic function. Greater impairment was found in subjects with a worse degree of hepatic fibrosis estimated by NFS. Hypoalbuminemia and thrombocytopenia were higher in subjects with a worse degree of hepatic fibrosis, whereas prevalence of SIBO positive was similar between the groups.

## Introduction

Non-alcoholic fatty liver disease (NAFLD) represents a spectrum of liver diseases characterized mainly by macro-vesicular steatosis that occurs in the absence of alcoholic consumption. The hepatic histology of patients with NAFLD varies from isolated hepatic steatosis to fatty liver with hepatocellular damage, inflammation and tissue fibrosis [1].

The prevalence of NAFLD and non-alcoholic steatohepatitis (NASH) are on the increase [2]. The progression of liver diseases has been described as an important cause for the development of liver fibrosis, cirrhosis and hepatocellular carcinoma [3,4], becoming the main indication for liver transplantation in the next two decades [5].

The pathophysiology of NAFLD is multifactorial and has been suggested to include an altered gut microbiota composition. Gut microbes are an endogenous source of ethanol, which may be delivered to the liver in a continuous fashion and promote steatosis and liver injury [6]. Some studies have reinforced the concept that small intestinal bacterial overgrowth (SIBO) plays an important role in the pathogenesis of NAFLD through endotoxin of bacteria and a tumor necrosis factor (TNF) as effective mediators [7,8].

NAFLD also exacerbates insulin resistance, dyslipidemia and causes the release of pro-inflammatory, profibrogenic and vasoactive mediators that can promote the development of cardiac complications, autonomic dysfunction as well as in the development and persistence of atrial fibrillation and other arrhythmias [9,10]. Furthermore, autonomic neuropathy can be associated with slowed motility, particularly in the stomach and proximal small intestine that promotes a propensity for small intestinal bacterial overgrowth (SIBO) which, in turn, can promote bacterial translocation and drive inflammation.

Heart rate variability (HRV) analysis is widely used to characterize the functions of the autonomic nervous system (ANS) [11] and has been proposed as a useful tool in identifying patients at risk for sudden cardiac death [10]. In fatty liver disease, cardiac and autonomic impairments appear to be dependent on the level of liver fat, metabolic dysfunction, inflammation and fibrosis staging, and, to a lesser extent, alcohol intake [12].

Despite previous studies showing that patients with NAFLD display impaired cardio-metabolic function and gut dysbiosis, it has not yet been demonstrated, to our understanding, whether the worsening of NAFLD accentuates cardio-metabolic dysfunction and gut dysbiosis. In this study, we evaluated the cardio-metabolic parameters and small intestinal bacterial overgrowth in patients with different degrees of hepatic fibrosis estimated by a noninvasive bio-marker panel. We hypothesized that cardiac autonomic dysfunction, metabolic impairment, and small intestinal bacterial overgrowth are more prevalent in patients with NAFLD with a high risk of advanced fibrosis.

## Methods

### Ethical aspects

This cross-sectional study was conducted in accordance with the Declaration of Helsinki. The protocol of the original study entitled "Evaluation of effectiveness of intervention with

probiotics in treatment of non-alcoholic fatty liver disease" was submitted and approved by the Research Ethics Committee of the Lauro Wanderley University Hospital, Federal University of Paraiba (Reference number 03871618.0.0000.5183) and all procedures were conducted in agreement with Resolution 466/2012 of the National Health Council and the International Declaration of Helsinki. All participants provided written informed consent.

## Subjects

Thirty healthy subjects (control group) and 48 patients aged 18–75 years, both genders, with Non-Alcoholic Fatty Liver Disease (defined by imaging or histology, and lack of secondary causes of hepatic fat accumulation) and belonging to the Gastroenterology/Hepatology outpatient clinic of Lauro Wanderley University Hospital between April and November 2019 were included in the study.

The diagnosis of NAFLD was performed by abdominal ultrasound and carried for physicians with experience in imaging diagnosis of liver steatosis. In addition, one subject had a liver biopsy confirming their NAFLD diagnosis. All subjects recruited for the study tested negative for viral serology (B and C hepatitis) and denied significant alcohol consumption. Significant alcohol consumption was defined as > 21 doses per week for men and > 14 doses per week for women over a 2-year period prior to inclusion in the study [13].

All participants underwent assessment of their autonomic functions and anthropometry measurements. The blood biochemistry measurements and hydrogen breath tests were performed only with the NAFLD subjects. Exclusion criteria consisted of a history consistent with pregnancy, cardiac transplant, presence of arrhythmias (e.g., ventricular atrial block, atrial fibrillation), cardiac pacemakers, ischemic and non-ischemic cardiomyopathy, important psychiatric diseases, active malignant neoplasms, evidence of other liver disease (autoimmune hepatitis, viral hepatitis, drug-induced liver injury, haemochromatosis, cholestatic liver disease or Wilson's disease) and history of alcohol consumption.

The participants were divided into three groups: healthy control (n = 30), NAFLD with low risk of advanced fibrosis (NAFLD-LRAF, n = 17) and NAFLD with a high risk of advanced fibrosis (NAFLD-HRAF, n = 31). The degree of hepatic fibrosis was estimated by NAFLD fibrosis score (NFS). The NFS is based on age, hyperglycemia, BMI, platelet count, albumin level, and AST/ALT ratio and has been recognized as a clinically useful tool for identifying advanced fibrosis in patients with NAFLD [13]. It has been demonstrated that a score less than -1.455 have 90% sensitivity and 60% specificity to exclude advanced fibrosis, whereas a score >0.676 have 67% sensitivity and 97% specificity to identify the presence of advanced fibrosis [13]. Considering the impossibility of staging liver fibrosis by biopsy into intermediate risk category (cutoff -1.455–0.676) and recognizing that the subjects of the intermediate and high risk are at higher risk for diabetes mellitus, cardiovascular diseases and overall mortality [14], we decided allocate the subjects of intermediate and high categories into a single category. Thus, cutoff less than -1.455 was used to classify subjects as low risk of advanced fibrosis, while cutoff greater than -1.455 was used to classify subjects as high risk advanced fibrosis. This was performed to access whether the worsening of nonalcoholic fatty liver disease could impair cardiometabolic function and intestinal bacterial overgrowth of those patients.

## Clinical and anthropometric measurements

A questionnaire was administered to all participants in order to retrieve information regarding their age, sex, physical activity, nutritional counselling, previous diseases, time of illness and use of medication. Body weight was measured to the nearest 0.1 kg using an electronic scale; Height was measured to the nearest 0.5 cm using a stadiometer (W200/50 A, Welmy). BMI

(kg/m2) was calculated as the weight divided by the square of the height. Waist circumference (cm) was measured at the level of the mid-point between the inferior margin of the last rib and the iliac crest measured horizontally using a constant tension tape with the patient standing.

## Blood sample and biochemical analyses

Fasting samples were analyzed for NAFLD-HRAF and NAFLD-LRAF. Blood samples were collected by a qualified nurse, from patients after a 12-hour fast, without strenuous exercise 24 hours before and without drinking alcohol 72 hours prior to collection. Plasma glucose, creatinine and triglycerides levels were determined by an automated enzymatic method; high-density lipoprotein cholesterol (HDL-c), low-density lipoprotein cholesterol (LDL-c) and total cholesterol, by direct colorimetric method; aspartate aminotransferase (AST), alanine aminotransferase (ALT) and gamma-glutamyl transferase (GGT) by colorimetric kinetic method; ferritin by the immunoturbidimetric assay; albumin by endpoint assay. Blood count and glycated hemoglobin tests were performed by the following methods, respectively: electronic determination in a Pentra 120 ABX device and high-performance liquid chromatography (HPLC).

## Blood pressure, electrocardiogram recording and heart rate variability analysis

All subjects abstained from intense physical activity 24 hours prior to examination. In addition, patients abstained from caffeinated beverage consumption or any stimulant drink 36 hours prior to blood pressure (BP) and electrocardiogram (ECG) recording. No alcohol intake was permitted 72 hours prior to the observations. The BP was measured in seated subjects after 3 min of rest, with a Welch Allyn sphygmomanometer with appropriate arm circumference cuff. Lastly, patients were asked to fast overnight for 12 hours before the ECG.

The recordings were performed in the morning (8 AM–11 AM) in a quiet room. After an initial stabilization period of 5 mins, subjects were instructed to remain silent, breathing normally at tidal volume, at rest and in the supine position. Following, ECG measurements were recorded for 10 mins. The ECG model 26T-LTS (ADinstruments®, Bella Vista, NSW, Australia) was used and recordings were made with the 5-electrode configuration through Lab-Chart® data acquisition software (ADinstruments®, Bella Vista, NSW, Australia).

ECG was set to a sampling rate of 1 kHz, range of 2 millivolts using a digital filter of 50 Hz (low pass). All data were exported and blindly analyzed by an independent trained researcher using LabChart 8 software. ECG recordings were processed by computer software (ECG analysis module for LabChart Pro; ADInstruments) for automatic detection of the R waves and beat-by-beat calculation of RR interval. For detection of ECG, 80 ms was used to determine typical QRS width and for R waves at least 300 ms apart.

For ECG analysis, 240 ms was used for the maximum PR and 240 ms for maximum RT. The heart rate (HR) and the following measures of HRV analysis were determined; 1) time-domain parameters: average R-R interval, standard deviation between the duration of RR intervals (SDRR), the square root of the mean of the sum of the squares of the successive differences between adjacent normal-to-normal beats (RMSSD), and the number of pairs of successive normal-to-normal beat intervals that differed by 50 ms (pRR50); 2) frequency-domain parameters: low frequency (LF) band (from 0.04 to 0.15 Hz) and high frequency (HF) band (from 0.15 to 0.40 Hz) and the LF/HF ratio, the power of each spectral component was calculated in normalized units (un); 3) Nonlinear parameters (SD1 and SD2). Poincaré scatters plots were constructed and investigated as a nonlinear tool, including the transverse axes (SD1, an indicator of parasympathetic activity) and the longitudinal axes (SD2 a function of sympathetic and vagal activity) [15].

### Hydrogen breath test assessment

Subjects in NAFLD-LRAF and NAFLD-HRAF, after fasting for 8–12 hours, underwent a glucose hydrogen breath test (GHBT) shortly after the ECG recording. This breath testing had been preceded by recommendations that the patients avoid the use of antibiotics for 4 weeks and avoid pro-motility agents and laxatives for at least 1 week, and avoid fermentable foods (e.g., complex carbohydrates) for a day before the breath test. In addition, during the breath test, patients were asked to avoid smoking and minimize physical exertion[16]. Subjects exhaled twice into a standardized apparatus at baseline, the mean value was taken as the basal breath hydrogen. A standardized glucose solution (50g of glucose/200ml of water) was then ingested and a repeat of the breath samples were then collected every 30 minutes for 120 minutes, with attention paid to minimization of variability. Breath samples were analyzed immediately for $H_2$ using a Gastrolyzer–Gastro + (Bedfont Scientific Ltd, Maidstone, UK). Breath tests were interpreted as positive for SIBO when fasting breath hydrogen concentration was $\geq$ 20 ppm or the hydrogen measured had a rise > 20ppm above the baseline value[7].

### Statistical analysis

Values are reported as mean (95% confidence interval) or % (n). AST, ALT, GGT and ferritin were log-transformed to normalize the data and presented as geometric means and confidence intervals. The variable age was analyzed using the one-way ANOVA test, then followed by the Tukey post-test. BP, biochemical parameters, ECG analysis and HRV measures were analyzed using one-way ANCOVA with correction for age and gender. Categorical data were analyzed by chi-square. The Pearson correlation coefficient (r) was used to explore the relationship between the clinical, anthropometric, NFS and laboratory variables obtained with the HRV parameters. The correlations obtained were classified as poor ($r \leq 0.20$), weak (0.21–0.40), moderate (0.41–0.60), good (0.61–0.80), and excellent (0.81–1.00). Statistical analyses were performed using SPSS 20.0 (IBM Corporation, Armonk, NY). Differences were considered significant when $p \leq 0.05$.

## Results

### General characteristics

Among the 78 study participants, the mean age was (±SD) 41.03 ± 14.72 years; 52.56% were women. Anthropometrical and clinical characteristics of participants are presented in **Table 1**. The age, weight and BMI of participants in the NFLD-HRAF group were significantly higher than in the other two groups. No differences were observed in the proportion of diabetes, dyslipidemia and hypertension, and drug treatment among the groups.

### Clinical assessments

**Table 2** shows the blood pressure, biochemistry and glucose hydrogen breath test data between groups. Systolic blood pressure was similar among the three groups, however, diastolic blood pressure was higher in the NAFLD—LRAF group when compared with the control group (*p* = 0.010).

Platelet, GGT and albumin serum levels were significantly lower in the NAFLD–HRAF compared to NAFLD–LRAF group. No differences were found in leukocytes, lipid profile, fasting glucose, AST, ALT, ferritin, serum creatinine and SIBO prevalence (**Table 2**).

### Assessment of ECG measures and heart rate variability

Electrocardiogram analysis and HRV in the time domain, frequency domain and nonlinear measurements are reported in **Table 3**. The ECG analyses revealed a wide QRS interval in

**Table 1. Anthropometrical and clinical characteristics of the study population.**

| VARIABLES | Control (n = 30) | NAFLD- LRAF (n = 17) | NAFLD—HRAF (n = 31) | *p*-value |
|---|---|---|---|---|
| Age (years) | 27.2 (25.1–29.2) | 43.1 (38.3–47.9)[†] | 53.3 (49.1–57.5)[* #] | <0.001 |
| Weight (kg) | 64.9 (61.3–68.6) | 82.4 (74.3–90.5)[†] | 93.7 (81.7–105.6)[*] | <0.001 |
| BMI (kg/m2) | 23.1 (21.8–24.4) | 29.3 (26.7–31.9)[†] | 35.2 (31.4–39.0)[* #] | <0.001 |
| Waist circumference (cm) | - | 103.1 ± 11.4 | 112.2 ± 18.8 | 0.076 |
| Gender (M/F) | 15/15 | 9/8 | 13/18 | 0.930 |
| Diabetes—% (n) | - | 17.6 (3) | 25.8 (8) | 0.476 |
| Dyslipidemia—% (n) | - | 29.4 (5) | 38.7 (12) | 0.753 |
| Hypertension—% (n) | - | 41.1 (7) | 51.6 (16) | 0.555 |
| Physical activity—% (n) | - | 23.5 (4) | 54.8 (17) | 0.066 |
| Nutritional counseling—% (n) | - | 11.8 (2) | 35.5 (11) | 0.099 |
| Medications | | | | |
| Antiglycemics | | | | |
| Biguanides—% (n) | - | 17.6 (3) | 19.4 (6) | 1.000 |
| SGLT-2 inhibitors—% (n) | - | 5.88 (1) | 6.45 (2) | 1.000 |
| Thiazolidineodiones—% (n) | - | 0 (0) | 0 (0) | - |
| DPP-4 inhibitors | - | 0 (0) | 0 (0) | - |
| GLP-1 analogs | - | 0 (0) | 0 (0) | - |
| Insulin | - | 5.88 (1) | 9.68 (3) | 1.000 |
| Thiazolidinediones—% (n) | - | 0 (0) | 0 (0) | - |
| Antihypertensives | | | | |
| ARBII—% (n) | - | 41.2 (7) | 22.6 (7) | 0.200 |
| ACE I inhibitors—% (n) | - | - | 19.4 (6) | 0.076 |
| βBlocker—% (n) | - | 11.8 (2) | 16.1 (5) | 1.000 |
| Calcium blockers—% (n) | - | 17.6 (3) | 9.68 (3) | 0.651 |
| Diuretics—% (n) | - | 11.8 (2) | 16.1 (5) | 1.000 |
| Lipid Lowering | | | | |
| Statin—% (n) | - | 5.88 (1) | 29.0 (9) | 0.074 |
| Fibrate—% (n) | - | 5.88 (1) | 0 (0) | - |

Data are expressed as mean (95% confidence interval) or % (n).

[*]$P<0.05$ NAFLD—HRAF *versus* Control group

[#]$P<0.05$ NAFLD—LRAF *versus* NAFLD—HRAF group

[†]$P<0.05$ NAFLD—LRAF *versus* Control group

patients with NAFLD when compared with the control group (p = 0.007), despite values still less than 120ms. Heart rate, P duration, RR, PR, QTc and JT intervals were similar among all three groups (**Table 3**).

Analyzing HRV in the time domain, SDRR, RMSSD and pRR50 were similar among the groups (**Table 3**). In frequency domain, HF oscillations was higher in the NAFLD-HRAF than in the NAFLD-LRAF ($p$ = 0.048) and the LF/HF ratio was lower in the NAFLD-HRAF than in subjects with NAFLD-LRAF ($p$ = 0.020). Lastly, the non-linear measures SD1 and SD2 were similar among groups (**Table 3**).

We correlated clinical parameters, anthropometric measures, NFS, HR, BP and laboratory parameters with measurements of HRV in the time domain, frequency and nonlinear measurements (**Table 4**). Age and BMI of subjects correlated positively with NFS and SBP and negatively with SDRR, RMSSD, pRR50 and SD2 (**Table 4**). There was a negative correlation between SBP and DBP SDRR, RMSSD, pRR50 and SD1, while HR did not correlate with HRV

**Table 2. Blood pressure and laboratorial characteristics among groups.**

| VARIABLES | Control (n = 30) | NAFLD- LRAF (n = 17) | NAFLD—HRAF (n = 31) | p-value |
|---|---|---|---|---|
| SBP (mmHg) | 114 (107–121) | 124 (117–130) | 125 (119–132) | 0.110 |
| DBP (mmHg) | 69 (64–74) | 80 (75–84)† | 74 (69–79) | 0.010 |
| Hemoglobin (g/dl) | - | 14.0 (13.4–14.7) | 14.0 (13.5–14.5) | 0.924 |
| Leukocytes ($10^9$/L) | - | 6.47 (5.65–7.28) | 6.54 (5.91–7.16) | 0.901 |
| Platelet ($10^9$/L) | - | 307.6 (273.2–342.1) | 232.3 (206.1–258.3) | 0.002 |
| Total cholesterol (mg/dL) | - | 200.1 (177.4–222.7) | 179.9 (163.3–196.5) | 0.174 |
| HDL-cholesterol (mg/dL) | - | 45.4 (38.9–52.0) | 40.3 (35.5–45.2) | 0.233 |
| LDL-cholesterol (mg/dL) | - | 131.1 (111.4–150.9) | 126.0 (111.6–140.5) | 0.688 |
| Triglycerides (mg/dL) | - | 165.7 (105.6–225.9) | 162.6 (118.5–206.6) | 0.935 |
| Fasting glucose (mg/dL) | - | 101.4 (78.9–123.9) | 101.3 (84.8–117.8) | 0.991 |
| HbA1c (%) | - | 6.38 (5.67–7.09) | 6.35 (5.79–6.90) | 0.944 |
| AST (IU/L)* | - | 21.3 (15.6–28.8) | 21.7 (17.3–27.1) | 0.922 |
| ALT (IU/L) * | - | 22.4 (15.3–33.1) | 18.9 (14.3–25.0) | 0.486 |
| GGT (IU/L) * | - | 69.3 (47.4–102.4) | 40.4 (30.6–53.5) | 0.034 |
| Albumin (g/dL) | - | 4.32 (4.15–4.50) | 4.06 (3.94–4.19) | 0.026 |
| Ferritin (ng/ml)* | - | 115.5 (70.1–192.4)* | 134.2 (92.7–196.2) | 0.644 |
| Serum creatinine (mg/dL) | - | 1.13 (0.75–1.51) | 0.84 (0.56–1.13) | 0.247 |
| SIBO on GHBT—% (n) | - | 11.8 (2) | 6.45 (2) | 0.607 |
| NFS | | -2.32 (-3.27 - -1.44) | 0.54 (-0.12–1.21) | <0.001 |

Data are expressed as mean (95% confidence interval). One-way ANCOVA adjusted for age and gender.

\* Geometric mean

†P<0.05 NAFLD—LRAF versus Control group

measures (**Table 4**). In the evaluation of the metabolic profile, triglycerides serum levels correlated positively with SBP and DBP, but did not correlate with any other parameter of HRV (**Table 4**). There was a negative correlation between fasting glucose and HbA1c with LF oscillation; and AST serum levels correlated negatively with SDRR, RMSSD and SD1 (**Table 4**).

## Discussion

In the present study, we observed that patients with NAFLD presented impairment in cardiac autonomic function when compared to healthy subjects and the greater impairment was seen in patients with a higher risk of advanced fibrosis as estimated by NFS. Given the high prevalence of NAFLD and its correlation with an increased risk of cardiovascular disease [17,18], an early cardiometabolic screen may be useful for prognostic and prevention of cardiometabolic complications for these patients.

Autonomic dysfunction has been related in NAFLD patients [19,20]. Here, we found that subjects with NAFLD and high risk of advanced fibrosis had a lower LF/HF ratio than NAFLD patients with low risk of advanced fibrosis. These findings suggest that the worsening of the NAFLD could accentuate the cardiac autonomic dysfunction of these patients.

Similarly, a recent study demonstrated that a fibrosis staging may also be an important key for estimating the degree of cardiac and autonomic dysfunction in NAFLD patients [12]. Other studies found the stage of liver fibrosis and steatosis in NAFLD to be related to the incidence of cardiovascular diseases [21], but there are few reports regarding the influence of the degree of fibrosis in autonomic functions based on HRV parameters.

In the time domain, SDRR, RMSSD and pRR50 indices; and non-linear parameters (SD1 and SD2) were similar among groups. However, an earlier study found a low HRV in the time

**Table 3. Comparison of ECG analysis and heart rate variability measures among the three groups studied.**

| VARIABLES | Control (n = 30) | NAFLD- LRAF (n = 17) | NAFLD—HRAF (n = 31) | p-value |
|---|---|---|---|---|
| | | *ECG analysis* | | |
| HR (bpm) | 68.9 (64.8–73.1) | 69.9 (65.5–73.3) | 68.5 (64.7–72.4) | 0.945 |
| RR interval (ms) | 886 (830–943) | 874 (820–928) | 896 (843–949) | 0.836 |
| PR interval (ms) | 158 (144–172) | 157 (144–171) | 164 (151–178) | 0.751 |
| P duration (ms) | 86 (70–102) | 85 (70–100) | 92 (78–107) | 0.753 |
| QRS interval (ms) | 64 (51–77) | 92 (79–104)[†] | 96 (84–109)[*] | 0.007 |
| QTc interval (ms) | 374 (355–392) | 384 (367–402) | 375 (357–392) | 0.600 |
| JT interval (ms) | 281 (258–303) | 268 (247–290) | 261 (240–282) | 0.583 |
| | | *HRV analysis* | | |
| SDRR (ms) | 51.6 (41.4–61.9) | 44.3 (34.5–54.1) | 46.9 (37.2–56.5) | 0.605 |
| RMSSD (ms) | 54.9 (41.9–67.8) | 40.3 (29.1–51.6) | 41.6 (30.3–52.8) | 0.265 |
| PRR50 (ms) | 26.8 (17.6–36.0) | 17.8 (9.7–25.8) | 20.8 (12.8–28.8) | 0.358 |
| LF (nu) | 41.1 (32.7–49.3) | 50.9 (42.9–58.9) | 39.6 (31.5–47.8) | 0.070 |
| HF (nu) | 54.0 (47.2–60.8) | 46.9 (40.4–53.4) | 57.5 (50.8–64.2)[#] | 0.048 |
| LF/HF | 0.83 (0.48–1.18) | 1.39 (1.06–1.73)[†] | 0.86 (0.52–1.20)[#] | 0.020 |
| SD1 (ms) | 35.3 (26.8–43.8) | 28.9 (20.8–37.0) | 30.2 (22.3–38.2) | 0.579 |
| SD2 (ms) | 62.5 (49.6–75.5) | 54.7 (42.3–67.0) | 57.9 (45.7–70.0) | 0.690 |

Data are expressed as mean (95% confidence interval). Control: health patients without diagnosis of NAFLD; NAFLD-LRAF: patients with NAFLD diagnosis and low risk of advanced fibrosis; NAFLD-HRAF: patients with NAFLD diagnosis and high risk of advanced fibrosis.

NAFLD: nonalcoholic fatty liver disease; HR: heart rate; SDRR: standard deviation of RR; RMSSD: square root of the mean squared differences of successive RR intervals; pRR50: number of pairs of successive normal-to-normal beat intervals that differed by 50 ms; LF nu: normalized unit in the low frequency band; HF nu: normalized unit in the high frequency band; SD: standard deviation of instantaneous RR interval variability.

[*] $P < 0.05$ NAFLD—HRAF *versus* Control group

[#] $P < 0.05$ NAFLD—LRAF *versus* NAFLD—HRAF group

[†] $P < 0.05$ NAFLD—LRAF *versus* Control group. One-way ANCOVA adjusted for age and gender.

domain in subjects with NAFLD and diabetes [22]. This suggests that the occurrence of diabetes in NAFLD subjects worsens autonomic dysfunction. An early study demonstrated that SD2 less than 25.5 ms and SDRR less than 20.4 ms may be used as significant predictors of mortality in cirrhotic patients [23]. Using these cutoffs, we can suggest that patients recruited in this cross-sectional study exhibited low risk of mortality.

Besides autonomic dysfunction and structural cardiac abnormalities, NAFLD is also associated with electrophysiological disorders of the heart [9,24]. QTc and QRS prolongation has been reported in patients with NAFLD [25,26]. The prolongation of these intervals is known to be arrhythmogenic and a predictor of cardiac mortality [27,28]. In our sample, we noticed higher significant values of QRS interval between individuals with steatosis when compared to the control group, despite values less than 120ms.

Low-grade systemic inflammation plays an important role in the impairment of cardiac function [29,30] and contributes to the progression of NASH [20]. The underlying mechanisms include expanded adipose tissue commonly observed in abdominal obesity and type 2 diabetes, lipotoxicity, oxidative stress and altered gut microbiota influenced by genetic and epigenetic variations [9].

Cardiovascular risk factors, such as obesity and diabetes, are associated with autonomic dysfunction and worsen the HRV parameters in NAFLD subjects [22,31]. In addition, other factors, such as gender and age, can influence on HRV [32]. Here, we found that age was positively correlated with SBP and negatively correlated with measures of SDRR, RMSSD, pRR50, LF and SD2. Thus, the results were corrected by the age and gender.

**Table 4. Linear correlation coefficients between heart rate variability measures and clinical, anthropometric, body composition and laboratory variables.**

| | NFS | SBP | DBP | HR | SDRR | RMSSD | pRR50 | LF | HF | LF/HF | SD1 | SD2 |
|---|---|---|---|---|---|---|---|---|---|---|---|---|
| Age (years) | 0.43* | 0.41* | 0.14 | -0.17 | -0.46* | -0.29* | -0.39* | -0.31* | 0.13 | -0.11 | -0.21 | -0.51* |
| BMI (kg/m2) | 0.34* | 0.32* | -0.10 | 0.08 | -0.15 | -0.23* | -0.21 | -0.02 | -0.02 | 0.08 | -0.15 | -0.13 |
| SBP (mmHg) | 0.13 | – | 0.66* | -0.12 | -0.21 | -0.16 | -0.23* | -0.21 | 0.10 | -0.13 | -0.16 | -0.22 |
| DBP (mmHg) | -0.16 | 0.66* | – | -0.06 | -0.27* | -0.28* | -0.27 | 0.03 | -0.08 | 0.03 | -0.30* | -0.20 |
| HR (bpm) | 0.07 | -0.12 | -0.06 | – | -0.13 | -0.18 | -0.14 | 0.01 | 0.05 | 0.05 | -0.18 | -0.08 |
| Hemoglobin (g/dl) | 0.03 | -0.04 | 0.37 | -0.01 | -0.12 | -0.24 | -0.16 | 0.12 | -0.14 | 0.12 | -0.25 | -0.07 |
| Leukocytes ($10^9$/L) | -0.02 | 0.15 | -0.12 | 0.07 | 0.20 | 0.14 | 0.16 | -0.19 | 0.14 | -0.10 | 0.09 | 0.23 |
| Platelet ($10^9$/L) | -0.41* | -0.12 | -0.26 | 0.12 | 0.12 | 0.14 | 0.12 | 0.05 | -0.02 | 0.10 | 0.08 | 0.09 |
| LDL-cholesterol (mg/dL) | -0.10 | 0.07 | -0.10 | 0.25 | 0.08 | 0.09 | 0.11 | -0.11 | 0.06 | -0.06 | 0.04 | 0.03 |
| HDL-cholesterol (mg/dL) | -0.20 | 0.10 | -0.28* | 0.16 | -0.06 | 0.07 | -0.09 | -0.02 | 0.03 | 0.04 | 0.12 | -0.16 |
| Triglycerides (mg/dL) | -0.04 | 0.25* | 0.35* | -0.07 | 0.09 | 0.05 | 0.10 | -0.28 | 0.26 | -0.25 | -0.00 | 0.10 |
| Fasting glucose (mg/dL) | 0.20 | 0.26 | 0.14 | -0.06 | -0.25 | -0.16 | -0.22 | -0.29* | 0.12 | -0.22 | -0.13 | -0.22 |
| HbA1c (%) | 0.14 | 0.27* | 0.08 | -0.01 | -0.22 | -0.17 | -0.21 | -0.31* | 0.16 | -0.25 | -0.12 | -0.19 |
| AST (IU/L) | 0.06 | -0.02 | 0.20 | 0.09 | -0.33* | -0.30* | -0.22 | -0.02 | -0.02 | 0.06 | -0.30* | -0.27 |
| ALT (IU/L) | -0.55* | 0.02 | 0.16 | 0.11 | -0.13 | -0.20 | -0.20 | 0.16 | -0.20 | 0.22 | -0.13 | -0.08 |
| GGT (IU/L) | -0.26 | 0.23 | 0.21 | -0.00 | 0.01 | 0.11 | -0.01 | -0.24 | 0.09 | -0.09 | 0.09 | -0.02 |
| Albumin (g/dL) | -0.26* | 0.11 | 0.32* | 0.22 | -0.07 | -0.08 | -0.05 | 0.12 | -0.15 | 0.21 | -0.04 | -0.07 |
| Ferritin (ng/ml) | -0.13 | 0.16 | 0.26 | 0.06 | 0.07 | 0.08 | 0.02 | -0.11 | 0.14 | -0.15 | 0.06 | 0.11 |
| NFS | – | 0.13 | -0.16 | 0.07 | -0.06 | -0.05 | -0.00 | -0.29* | 0.27 | -0.30* | -0.04 | -0.04 |

*p<0.05, Pearson correlation coefficient (R).

Abbreviations: AST, aspartate aminotransferase; ALT, alanine aminotransferase; BMI, body mass index; LDL-cholesterol, low density lipoprotein cholesterol; GGT, and gamma-glutamyl transferase; HDL-cholesterol, high density lipoprotein cholesterol; HbA1c, glycated hemoglobin; SBP, systolic blood pressure; DBP, diastolic blood pressure; HR (bpm), heart rate; NFS, non-alcoholic fatty liver disease score; SDRR (ms), standard deviation of RR; RMSSD (ms), square root of the mean squared differences of successive RR interval; pRR50 (ms), number of pairs of successive normal-to-normal beat intervals that differed by 50 ms; LF nu, normalized unit in the low frequency band; HF nu, normalized unit in the high frequency band; SD1/SD2- SD, standard deviation of instantaneous RR interval variability.

Glucose and Lipid dysmetabolism have been associated with impaired HRV and cardiac autonomic dysfunction in obese and NAFLD subjects [15,22]. However, in the present study, the glucose and lipid serum levels did not correlate expressively with worsening of HRV indices; AST serum levels, however, were negatively correlated with SDRR, RMSSD and SD1. This reveals that HRV indices may be influenced by the various clinical and metabolic conditions.

It has been to suggested that patients with NAFLD have an increased gut permeability or gut dysbiosis, which may lead to enhanced leakage of endotoxins (especially, lipopolysaccharide—LPS) from the gut, increased production of proinflammatory cytokines and autonomic disturbance [33–35].

Small intestinal bacterial overgrowth (SIBO) is characterized by an excessive number of bacteria in jejunal contents and a significant association with NAFLD has been reported [36,37]. A greater prevalence of SIBO in NAFLD patients has been reported by many authors [7,35]. In the present study, only four (8.33%) NAFLD subjects were SIBO positive and SIBO prevalence was similar between NAFLD-LRAF and NAFLD-HRAF subjects. Although the prevalence observed it was higher to that reported in another Brazilian study [38], it is notably lower than seem in studies conducted in other countries. For example, 39% SIBO positive was found in NAFLD patients from Iran, 60% SIBO positive was found in Italy and 31.3% SIBO positive was found in Indonesia [7,35].

In most cases of NAFLD, except in patients with associated cirrhosis, liver function markers are normal. Ferritin levels are elevated in more than 20% of these patients [39]. Liver enzyme

levels are within the normal range in approximately two-thirds of NAFLD patients [40]. In addition, there is a close association between NAFLD and metabolic risk factors, particularly dyslipidemia and type 2 diabetes mellitus [41].

In present study, although platelets count, albumin and GGT concentrations were reduced in NAFLD-HRAF than NAFLD-LRAF subjects, glyco-lipid and hepatic enzymes were similar between groups. Despite this, a negative correlation was found between AST and glycemic variables and HRV indices, demonstrating the importance of the biochemical measurement for evaluate the possibility of cardiovascular complications in NAFLD subjects.

## Potential limitations

Our study was conducted with a specific population and generalizability or transportability might not apply. We have used abdominal imaging as a diagnostic method for NAFLD, and NFS as a predictor of fibrosis, because liver biopsies were not available for this study. In addition, we did not use gut aspiration techniques to assess gut dysbiosis, although this is considered the gold standard.

## Conclusion

NAFLD impairs cardiac autonomic function and may cause greater impairment in the sympathovagal balance as noted in individuals with a worse degree of hepatic fibrosis estimated by NFS. The presence of SIBO was not associated to the degree of NAFLD in the same way as biochemical metabolic parameters were.

## Author Contributions

**Conceptualization:** Marília Marques Pereira Lira, José Luiz de Brito Alves.

**Data curation:** Marília Marques Pereira Lira, Vinícius José Baccin Martins, Gitana da Silva.

**Formal analysis:** Vinícius José Baccin Martins.

**Investigation:** Marília Marques Pereira Lira, Gitana da Silva, Éder Jackson Bezerra de Almeida Filho.

**Supervision:** José Eymard Moraes de Medeiros Filho, José Luiz de Brito Alves.

**Writing – original draft:** Marília Marques Pereira Lira, José Luiz de Brito Alves.

**Writing – review & editing:** José Eymard Moraes de Medeiros Filho, Vinícius José Baccin Martins, Francisco Antônio de Oliveira Junior, Alexandre Sérgio Silva, João Henrique da Costa-Silva, José Luiz de Brito Alves.

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
