## [Decision Letter · Decision Letter 0]

16 Jul 2020

PONE-D-20-12922

Association of worsening of nonalcoholic fatty liver disease with cardiometabolic function and intestinal bacterial overgrowth: a cross-sectional study

PLOS ONE

Dear Dr. Brito Alves,

Thank you for submitting your manuscript to PLOS ONE. After careful consideration, we feel that it has merit but does not fully meet PLOS ONE’s publication criteria as it currently stands. Therefore, we invite you to submit a revised version of the manuscript that addresses the points raised during the review process.

There is a number of issues that have to be addressed before the manuscript becomes acceptable. Please make sure that all the issues raised by the reviewer are properly addressed in the revised version.

We look forward to receiving your revised manuscript.

Kind regards,

Oscar Millet

Academic Editor

PLOS ONE

Journal Requirements:

"The funders had no role in study design, data collection and analysis, decision to

publish, or preparation of the manuscript."

Reviewers' comments:

Reviewer's Responses to Questions

**Comments to the Author**

1. Is the manuscript technically sound, and do the data support the conclusions?

Reviewer #1: Yes

2. Has the statistical analysis been performed appropriately and rigorously? 

Reviewer #1: Yes

3. Have the authors made all data underlying the findings in their manuscript fully available?

Reviewer #1: No

4. Is the manuscript presented in an intelligible fashion and written in standard English?

Reviewer #1: Yes

5. Review Comments to the Author

Reviewer #1: The authors present a study whose main objective is to test if worsening of NAFLD is associated with cardiac dysfunction, intestinal bacterial overgrowth or metabolic impairment. They considered the risk of advanced fibrosis as worsening criteria for NAFLD. This risk was measured with a known noninvasive method (NFS), which was applied only to NAFLD patients that were diagnosed by abdominal imaging. The NAFLD cohort was complemented with a small control cohort of healthy subjects. For both cohorts, they obtained anthropometric parameters, blood pressure and ECG analysis. For NAFLD cohort, also biochemical analysis and hydrogen breath tests were performed. They did not find differences of bacterial overgrowth regarding advanced fibrosis risk. Biochemical analysis showed some differences for platelets, albumin and GGT. In the case of heart variability measures, they only found differences between low risk and high risk for HF parameter and LF/HF ratio.

In my opinion, it is a well-explained, honest and rigorous study whose results are of interest for scientific community. Although the study found a small set of differences between low risk and high risk of advanced fibrosis, also the lack of differences in other parameters is itself a result. Therefore, I consider this manuscript is acceptable for publication with minor revisions:

1) Line 106: “Non-Alcoholic Fatty Liver Disease (defined by imaging or histology, and lack of secondary causes of hepatic fat accumulation)”. Could you provide more information about the specific criteria to diagnose NAFLD?

2) Could you explain why -1,455 is used as NFS cutoff value for high risk of advanced fibrosis?

3) There is a big age difference between control group and NAFLD groups. Could you explain why did you select a control group younger than NAFLD group?

6. PLOS authors have the option to publish the peer review history of their article (what does this mean?). If published, this will include your full peer review and any attached files.

Reviewer #1: No

---

## [Author Response · Author response to Decision Letter 0]

20 Jul 2020

Reviewer #1: The authors present a study whose main objective is to test if worsening of NAFLD is associated with cardiac dysfunction, intestinal bacterial overgrowth or metabolic impairment. They considered the risk of advanced fibrosis as worsening criteria for NAFLD. This risk was measured with a known noninvasive method (NFS), which was applied only to NAFLD patients that were diagnosed by abdominal imaging. The NAFLD cohort was complemented with a small control cohort of healthy subjects. For both cohorts, they obtained anthropometric parameters, blood pressure and ECG analysis. For NAFLD cohort, also biochemical analysis and hydrogen breath tests were performed. They did not find differences of bacterial overgrowth regarding advanced fibrosis risk. Biochemical analysis showed some differences for platelets, albumin and GGT. In the case of heart variability measures, they only found differences between low risk and high risk for HF parameter and LF/HF ratio.

In my opinion, it is a well-explained, honest and rigorous study whose results are of interest for scientific community. Although the study found a small set of differences between low risk and high risk of advanced fibrosis, also the lack of differences in other parameters is itself a result. Therefore, I consider this manuscript is acceptable for publication with minor revisions:

Answer: We would like to thank the reviewer 1 for recognizing the relevance of our work and the opportunity to improve our manuscript.

1) Line 106: “Non-Alcoholic Fatty Liver Disease (defined by imaging or histology, and lack of secondary causes of hepatic fat accumulation)”. Could you provide more information about the specific criteria to diagnose NAFLD?

Answer: We appreciate this observation. For the NAFLD diagnosis we used abdominal ultrasound, carried out by specialist, as a first line diagnostic method. We considered mild hepatic steatosis or grade 1 when a slight and diffuse increase in hepatic echogenicity was noticed with normal visualization of the intrahepatic vessels and the diaphragm. Moderate or grade 2 hepatic steatosis was defined by a moderate and diffuse increase in echogenicity and blurring in the visualization of intrahepatic vessels and the diaphragm. Marked hepatic steatosis or grade 3 was diagnosed by a marked increase in echogenicity of the liver parenchyma associated with poor or non-visualization of intrahepatic vessels, diaphragm and posterior region of the liver. 

In revised version, we have added the sentence: “The diagnosis of NAFLD was performed by abdominal ultrasound and carried for physicians with experience in imaging diagnosis of liver steatosis. In addition, one subject had a liver biopsy confirming their NAFLD diagnosis. All subjects recruited for the study tested negative for viral serology (B and C hepatitis) and denied significant alcohol consumption. Significant alcohol consumption was defined as > 21 doses per week for men and > 14 doses per week for women over a 2-year period prior to inclusion in the study [13]. 

2) Could you explain why -1,455 is used as NFS cutoff value for high risk of advanced fibrosis?

Answer: We would like to thank the reviewer 1 for this observation. In revised version, we have explained the reason for use -1,455 as NFS cutoff value for high risk of advanced fibrosis. In pag 6, lines 129-143, we have added: “The NFS is based on age, hyperglycemia, BMI, platelet count, albumin level, and AST/ALT ratio and has been recognized as a clinically useful tool for identifying advanced fibrosis in patients with NAFLD [13]. It has been demonstrated that a score less than -1.455 have 90% sensitivity and 60% specificity to exclude advanced fibrosis, whereas a score >0.676 have 67% sensitivity and 97% specificity to identify the presence of advanced fibrosis [13]. Considering the impossibility of staging liver fibrosis by biopsy into intermediate risk category (cutoff -1.455 – 0.676) and recognizing that the subjects of the intermediate and high risk are at higher risk for diabetes mellitus, cardiovascular diseases and overall mortality [14], we decided allocate the subjects of intermediate and high categories into a single category. Thus, cutoff less than -1.455 was used to classify subjects as low risk of advanced fibrosis, while cutoff greater than -1.455 was used to classify subjects as high risk advanced fibrosis. This was performed to access whether the worsening of nonalcoholic fatty liver disease could impair cardiometabolic function and intestinal bacterial overgrowth of those patients.”

3) There is a big age difference between control group and NAFLD groups. Could you explain why did you select a control group younger than NAFLD group?

Answer: We would like to thank the reviewer 1 for this observation. Previous studies have demonstrated that HRV change with age and gender. Then, our main reason when we performed the recruitment of patients was try to reduce the effect of age on HRV measurements and have results that reflected the HRV parameters in a healthy population and not provoked by advancing age. Lastly, for greater control of the results obtained in present study, we performed a one-way ANCOVA test with correction for age and gender. This gives us a more control to affirm that our findings are not being modulated by age or gender.

---

## [Editor Report · Decision Letter 1]

24 Jul 2020

Association of worsening of nonalcoholic fatty liver disease with cardiometabolic function and intestinal bacterial overgrowth: a cross-sectional study

PONE-D-20-12922R1

Dear Dr. Brito Alves,

We’re pleased to inform you that your manuscript has been judged scientifically suitable for publication and will be formally accepted for publication once it meets all outstanding technical requirements.

Kind regards,

Oscar Millet

Academic Editor

PLOS ONE
---

## [Editor Report · Acceptance letter]

17 Aug 2020

PONE-D-20-12922R1 

Association of worsening of nonalcoholic fatty liver disease with cardiometabolic function and intestinal bacterial overgrowth: a cross-sectional study 

Dear Dr. de Brito Alves:

I'm pleased to inform you that your manuscript has been deemed suitable for publication in PLOS ONE. Congratulations! Your manuscript is now with our production department. 

Kind regards, 

on behalf of

Dr. Oscar Millet 

Academic Editor

PLOS ONE